# Salivary Biomarkers in COVID-19 Patients: Towards a Wide-Scale Test for Monitoring Disease Activity

**DOI:** 10.3390/jpm11050385

**Published:** 2021-05-08

**Authors:** Cecilia Napodano, Cinzia Callà, Antonella Fiorita, Mariapaola Marino, Eleonora Taddei, Tiziana Di Cesare, Giulio Cesare Passali, Riccardo Di Santo, Annunziata Stefanile, Massimo Fantoni, Andrea Urbani, Gaetano Paludetti, Gian Ludovico Rapaccini, Gabriele Ciasca, Umberto Basile

**Affiliations:** 1Dipartimento di Scienze Mediche e Chirurgiche, UOC Gastroenterologia, Fondazione Policlinico Universitario “A. Gemelli” IRCCS, Università Cattolica del Sacro Cuore, 00168 Rome, Italy; cecilia.napodano@gmail.com (C.N.); gianludovico.rapaccini@policlinicogemelli.it (G.L.R.); 2Dipartimento di Scienze di Laboratorio e Infettivologiche, Fondazione Policlinico Universitario “A. Gemelli” IRCCS, Università Cattolica del Sacro Cuore, 00168 Rome, Italy; cinziaanna.calla@unicatt.it (C.C.); stefanile.nunzia@gmail.com (A.S.); andrea.urbani@policlinicogemelli.it (A.U.); 3Dipartimento di Scienze dell’Invecchiamento, Neurologiche, Ortopediche e della Testa e del Collo, Fondazione Policlinico Universitario “A. Gemelli” IRCCS, Università Cattolica del Sacro Cuore, 00168 Rome, Italy; antonella.fiorita@policlinicogemelli.it (A.F.); tizianadicesare90@gmail.com (T.D.C.); giuliocesare.passali@policlinicogemelli.it (G.C.P.); gaetano.paludetti@policlinicogemelli.it (G.P.); 4Dipartimento di Medicina e Chirurgia Traslazionale, Fondazione Policlinico Universitario “A. Gemelli” IRCCS, Università Cattolica del Sacro Cuore, 00168 Rome, Italy; mariapaola.marino@unicatt.it; 5Dipartimento di Sicurezza e Bioetica, Sezione di Malattie Infettive, Fondazione Policlinico Universitario “A. Gemelli” IRCCS, Università Cattolica del Sacro Cuore, 00168 Rome, Italy; eleonora.taddei@policlinicogemelli.it (E.T.); massimo.fantoni@policlinicogemelli.it (M.F.); 6Dipartimento di Neuroscienze, Sezione di Fisica, Fondazione Policlinico A. Gemelli IRCCS, Università Cattolica del Sacro Cuore, 00168 Rome, Italy; riccardo.disanto92@gmail.com (R.D.S.); gabriele.ciasca@gmail.com (G.C.)

**Keywords:** COVID-19, IgA, FLC, salivary biomarkers

## Abstract

The ongoing outbreak of coronavirus disease 2019 (COVID-19), which impairs the functionality of several organs, represents a major threat to human health. One of the hardest challenges in the fight against COVID-19 is the development of wide-scale, effective, and rapid laboratory tests to control disease severity, progression, and possible sudden worsening. Monitoring patients in real-time is highly demanded in this pandemic era when physicians need reliable and quantitative tools to prioritize patients’ access to intensive care departments. In this regard, salivary biomarkers are extremely promising, as they allow for the fast and non-invasive collection of specimens and can be repeated multiple times. Methods: We compare salivary levels of immunoglobulin A subclasses (IgA1 and IgA2) and free light chains (kFLC and λFLC) in a cohort of 29 SARS-CoV-2 patients and 21 healthy subjects. Results: We found that each biomarker differs significantly between the two groups, with *p*-values ranging from 10^−8^ to 10^−4^. A Receiving Operator Curve analysis shows that λFLC level is the best-suited candidate to discriminate the two groups (AUC = 0.96), with an accuracy of 0.94 (0.87–1.00 95% CI), a precision of 0.91 (0.81–1.00 95% CI), a sensitivity of 1.00 (0.96–1.00 95% CI), and a specificity of 0.86 (0.70–1.00 95% CI). Conclusion: These results suggest λFLC as an ideal indicator of patient conditions. This hypothesis is strengthened by the consideration that the λFLC half-life (approximately 6 h) is significantly shorter than the IgA one (21 days), thus confirming the potential of λFLC for effectively monitoring patients’ fluctuation in real-time.

## 1. Introduction

The severe acute respiratory syndrome coronavirus 2 (SARS-CoV-2) emerged in late 2019 in the city of Wuhan, China, causing an outbreak of unusual viral pneumonia named ’coronavirus disease 2019′ (COVID-19) which threatens human health and public safety [1]. Being highly transmissible, this novel coronavirus disease spread dramatically all over the world. Due to the wide range of organ and tissue impairments it causes, paralleled by abnormalities in inflammatory markers, COVID-19 could be considered a systemic pathology [2,3].

In these challenging times, there is an urgent need to develop broad-scale population-based tests to improve COVID-19 prevention and diagnosis, as well as to control outcomes and monitor herd immunity. In this context, saliva is a promising candidate as it allows for the detection of SARS-CoV-2 RNA and anti-SARS-CoV-2 antibodies [2]. Additionally, saliva collection is relatively easy and non-invasive, does not require specialist training, and tends to be preferable to blood collection when repeated sampling is required.

Salivary immunoglobulin A (IgA) represents an important immune biomarker secreted by healthy people and patients in different clinical conditions, or in response to acute mental stress and exercise [4]. In the oral cavity, it has a key role among mucosal defense proteins, being a fundamental player in the first line of defense against pathogen adhesion and penetration into tissues [3,5]. Moreover, the occurrence of autoantibodies of the IgA class in several autoimmune diseases suggests that these molecules have more complex functions in regulating the immune response [6]. Two classes of IgAs can be distinguished: IgA1 and IgA2. These are not equally distributed within the humoral and mucosal immune systems. IgA1 is the major (approximately 80%) subclass in serum, while IgA2 is the major component in secretions [7]. The majority of IgA is secreted as a dimer into mucosal tissues where it regulates immune homeostasis, while the monomeric form in serum can mediate pro-inflammatory responses through the release of cytokines and chemokines and the activation of phagocytosis and degranulation. Furthermore, IgA subclasses act differently on neutrophils and macrophages, with IgA2 promoting inflammatory reactions through neutrophil extracellular trap formation and inflammatory cytokine expression [8]. Recently, subclass-specific antisera have been applied in a turbidimetric assay to establish age-dependent reference values for serum concentrations of the two IgA subclasses in children and adults [7]. The biological variability of IgA makes its interpretation a challenging task, pointing out the need for its inclusion into more articulated diagnostic, monitoring, and follow-up algorithms of immunological diseases [9,10].

The serological free light chains (FLC) of immunoglobulins are attracting a lot of attention as novel biomarkers of inflammatory diseases. These molecules are produced in excess by plasma cells during immunoglobulin synthesis and released into circulation, where they should be considered bioactive molecules, rather than a byproduct without any functional relevance. FLCs’ role in the course of autoimmune diseases has been deeply investigated [11,12,13,14].

This study aims to compare levels of salivary IgA subclasses and FLCs in COVID-19 patients and control subjects, evaluating the activation and involvement of oral immunity during the disease. The salivary markers’ diagnostic performance was also tested. Thanks to the easy accessibility of saliva, these immunological markers can represent a useful and convenient tool for assessing the activity of the disease and monitoring it over time.

## 2. Materials and Methods

### 2.1. Patients and Controls

We enrolled 29 COVID-19 patients (17 males and 12 females, mean age 49.7 years, SD 10.6 years) for this monocentric prospective study among patients admitted to the COVID-19 Unit at Fondazione Policlinico Universitario “A.Gemelli”, IRCCS, Rome; 21 healthy donators (HD) (10 males and 11 females, mean age 44.7 years, SD 11.6 years) served as a control group. Age values between the two groups were compared with a Wilcoxon test (*p* = 0.19), showing the absence of statistically significant differences.

Salivary specimens were collected from May 2020 to July 2020. The whole patient population consisted of hospitalized COVID-19 adult patients with a positive molecular assay test result for SARS-CoV-2 and characteristic symptoms of COVID-19: fever (temperature greater than 37.5 °C), dry cough, and pharyngodynia. HDs were enrolled from the health personnel of our institution. They had nasopharyngeal swabs negative for the presence of SARS-CoV-2 virus, performed no more than 24 h before serum and saliva collection. They were also negative for anti-SARS-CoV-2 antibodies in the serum. Patients and HDs under the age of 18 were not included. Other exclusion criteria were the presence of plasma cell disorders, autoimmune disease, the presence of autoantibody, antinuclear antibody positivity, the presence of a monoclonal component in capillaries electrophoresis, and the presence of chronic hepatitis (C, B). The clinical and demographic characteristics of the enrolled patients are reported in Table 1.

Patients presented with the following comorbidities: diabetes (*n =* 3), hypertension (*n =* 6), obesity (*n =* 4), oncology history (*n =* 2), and heart disease (*n =* 4). Symptoms at onset: fever (*n =* 22), cough (*n =* 7), dyspnea (*n =* 7), and asthenia (*n =* 10). Several complications were experienced: 21 patients had bilateral pneumonia, 11 patients were treated with oxygen therapy, and 6 patients were smokers. Salivary immune response biomarkers including FLCs and IgA and IgA subclasses were measured in both groups and are summarized in Table 2.

### 2.2. Laboratory Testing

In all subjects, one sample of serum and one sample of saliva were collected. The study was cross-sectional, and none of the subjects contributed more than one saliva–serum paired sample.

Blood was collected by venipuncture into plain vacutainer tubes. Whole blood samples were allowed to clot at room temperature for 20 min and then centrifuged at 3000× *g* for 10 min at 4 °C; separated serum was immediately stored at −80 °C until analysis.

Unstimulated saliva samples were collected with participants in the seated position, leaning forward with their head tilted forward. For saliva sampling, patients and healthy controls were provided with the following instructions before samples were taken: Avoid the consumption of solid foods one hour before the test and perform oral and dental hygiene half an hour before the evaluation. Sampling was performed between 09:00 h and 11:00 h to minimize the variations associated with the circadian cycle. Saliva collections lasted 3 min, during which time participants had a Salivette^®^ placed in their mouth behind their teeth, which was subsequently centrifuged at 10,000× *g* for 2 min to separate cells and insoluble matter. The supernatant was then removed and stored at −80 °C until assay.

Saliva samples were tested for salivary IgA subclasses and free k and λ chains. IgA subclasses were assessed by Optilite analyzer (Optlite IgA CSF kit, Optilite IgA1, and IgA2 subclasses kit, The Binding Site, UK; salivary IgA normal range: 8.7–20 mg/dl). Total IgA levels were also measured in serum samples in 20 out of 29 patients (serum normal range for IgA: 0.845–4.990 g/L; for IgA1: 760.81–3282.03 mg/L; for IgA2: 68.9–1142.5 mg/L).

FLCs were measured using the Optilite analyzer (Freelite™ Human Kappa and Lambda Free Kits, The Binding Site, UK; serum normal range: 3.3–19.4 mg/L for free k and 5.7–26.3 mg/L for free λ). A ratio of k/λ < 0.26 or >1.65 was considered abnormal.

Samples were thawed only once and immediately assayed in a single batch, following the manufacturer’s instructions. Each sample was tested twice to minimize eventual discrepancies, and all tests were performed in the same laboratory with the same instruments.

### 2.3. Ethical Consideration

This study complied with the Ethical Principles for Medical Research Involving Human Subjects in accordance with the World Medical Association Declaration of Helsinki and was certified by the Committee of the Applicable Institution of the Fondazione Policlinico Universitario Agostino Gemelli IRCCS (FPG), Rome.

The ethics committee of the institution (FPG) approved the study (ID: 3222). All patients gave written informed consent to the use of their clinical and serological data in this study. The whole study was conducted according to the Declaration of Helsinki, as revised in 2013.

### 2.4. Statistical Analysis

Statistical analyses were performed using the software package R (4.0.2 release, R Foundation, Vienna) [15]. Laboratory parameters were tested for normality using a visual inspection of the Q–Q plot, followed by a Shapiro–Wilk test, and showed significant deviations from normality. Comparisons between groups were performed with the Wilcoxon Unpaired Two-Sample Test. The diagnostic performance of the investigated salivary markers in distinguishing the two groups was assessed by logistic regression followed by ROC curve analysis.

Logistic regression was performed using the glm function from the R package ‘stats’ to extract probabilities from the fitted models, either with a single biomarker or with several biomarkers used in combination. ROC curves and AUC values were calculated as previously described, using the R package ‘pROC’ [16,17,18]. Stepwise backward logistic regression was used to select the best subset of k parameters that minimizes the Akaike’s Information Criterion (AIC) = 2k−2 ln (L), where L is the maximum value of the likelihood function for the model [19]. A Bayesian information criterion (BIC) was also tested, and similar results were obtained (data not shown). The classifiers were evaluated with a bootstrap procedure carried out on the logistic regression, which allowed us to find out the distribution of performance indicators such as accuracy, precision, sensitivity, and specificity. For this purpose, we performed 5000 bootstrap samples of the investigated datasets, and we used them to compute the logistic regression parameters with the corresponding confusion matrix. Ultimately, the ensemble of generated confusion matrixes was used to derive the corresponding distribution of the desired statistics.

Correlations between variables were evaluated with Spearman’s correlation coefficients. Strength of correlation was evaluated, considering coefficients >0.68 as strong correlation, 0.35–0.68 as moderate, and <0.35 as weak correlation [20]. Correlation heat maps were calculated with the package ‘corrplot’ implemented in the software R, according to [21,22].

## 3. Results

In Figure 1A–F, we show a box plot analysis of the selected salivary markers measuring subjects’ immune response together with the result of a Wilcoxon Unpaired Two-Sample Test. Data are represented using a logarithmic scale and summarized in Table 2. A statistically significant increase in salivary levels in COVID-19 patients compared to healthy controls was detected for all the investigated biomarkers, namely kFLC (*p* = 1.8 × 10^−4^, Figure 1A), λFLC (*p* = 1.4 × 10^−8^, Figure 1B), k/λ (*p* = 1.3 × 10^−2^, Figure 1C), k + λ (*p* = 6.6 × 10^−5^, Figure 1D), IgA1 (*p* = 1.3 × 10^−5^, Figure 1E), and IgA2 (*p* = 2.0 × 10^−5^, Figure 1F).

ROC curves and AUC values were computed to assess the performance of salivary biomarkers in distinguishing the two groups. A remarkable increase in the ROC curves for low values of the *x*-axis can be observed in Figure 2A for all the investigated biomarkers, confirming their diagnostic potential. As expected from Figure 2A, large and significant AUC values were measured for each marker (Table 3). We attempted to improve the effectiveness of the model shown in Figure 2B by performing a stepwise logistic regression including all the measured markers. To this end, first the complete model was obtained and then a stepwise backward selection was performed to highlight the most relevant subset of parameters that minimizes the Akaike’s Information Criterion (AIC).

The computed AIC value for each cycle of the procedure is reported in Figure 2B, showing the progressive removal of less informative biomarkers. The minimization procedure selected λ and IgA2 and discarded the other parameters. In Figure 2C, we show the computed ROC curve for the step model, based on the two selected variables. All of the computed AUC values, together with the corresponding 95% CIs, are compared in Figure 2D. An analysis of this figure suggests that the combined use of λ and IgA2 does not provide significant improvement compared to λ alone. A more in-depth comparison of the two models was performed using a bootstrap procedure aimed at estimating the statistical distribution of selected performance parameters, including accuracy, precision, sensitivity, and specificity. Bootstrap implementation is explained in the statistical analysis section. In Figure 3, we show the computed distribution of accuracy (A,E), precision (B,F), sensitivity (C,G), and specificity (D,H) for λ (upper panels) and λ + IgA2 (lower panels). Despite both models showing very good classification abilities, the computed distributions appear to be very similar. Therefore, to keep the number of parameters in the model as low as possible, we conclude that the use of λ alone is preferable over the step model in Figure 2C with the available sample size. For this model, the following parameters (95% CIs) can be derived: accuracy = 0.94 (0.87–1.00), precision = 0.91 (0.81–1.00), sensitivity = 1.00 (0.96–1.00), and specificity = 0.86 (0.70–1.00). Mean parameters and confidence intervals were derived, fitting the retrieved distribution with a Gaussian curve (Appendix A).

To investigate in greater depth the relationship between immune response salivary markers and inflammatory blood markers in pathological subjects, we performed a correlational analysis. For this purpose, Spearman’s correlation coefficients were computed and visualized as a correlation map (Figure 4).

Significant Spearman’s correlations are summarized in Table 4, together with the corresponding *p*-values. As shown in Table 4 and Figure 4, correlations among salivary markers and between selected salivary and blood markers are strong and positive, thus hinting at a global and coordinated response of the immune system and inflammatory process during COVID-19 infection. Notably, ferritin levels appear to be negatively correlated with IgA2 levels, a result that deserves a better, in-depth study.

The potential effects on the levels of the salivary and plasmatic biomarkers of comorbidities, symptoms at onset, and complications are displayed in Appendix A for patients, showing the absence of statistically significant differences, with the following exceptions: patients diagnosed with diabetes or hypertension showed a significantly decreased k/λ ratio; patients with oncology history or undergoing oxygen therapy showed a significantly increased k/λ ratio. Concerning diabetes, results in Appendix A are in close agreement with those recently published by Matsumori and collaborators [23], which measured a decrease in the k/λ ratio on blood samples obtained from a cohort of diabetic subjects compared to controls. The authors hypothesize the involvement of NF-kB activation, which might alter the production of kFLC and λFLC. Moreover, we noticed a strong and expected association between diabetes and hypertension in our database (all diabetic subjects also suffer from hypertension), which is likely to be the reason for the decrease of the k/λ ratio in both populations.

## 4. Discussion

We hypothesize that oral mucosal surfaces, as the first line of defense against pathogens, are possible sites for initial triggering events of COVID-19. IgA antibodies are an essential component of the adaptive immune system’s ability to fight pathogens at the lining of mucosal surfaces and are capable of both pro- and anti-inflammatory effects [24,25]. The two subclasses, IgA1 and IgA2, display different levels in serum and mucosal fluids [8]. There is a large and increasing interest in researching salivary markers for SARS-Cov-2 diagnosis and monitoring. In this context, COVID-19 patients’ IgA levels in saliva were recently evaluated in two clinical papers [26,27]. To the best of our knowledge, this is the first study to date aimed at evaluating salivary IgA1 and IgA2 subclasses, as well as salivary FLC levels in COVID-19 patients in comparison to unvaccinated healthy subjects. Notably, we found a statistically significant increase in all of these biomarkers in the COVID-19 group when compared to HDs.

Additionally, we analyzed the overall amount of FLCs (k + λ) and the FLC k/λ ratio, which were significantly different between the two groups. We measured a very significant increase associated with remarkably low *p*-values that range from 10^−8^ to 10^−4^, except for k/λ (*p* = 0.013). The strength of this increase is even clearer considering that the data in Figure 1 are represented on a logarithmic scale, due to the large data range, and that there is little or no superposition between interquartile ranges. These findings suggest investigating the possible introduction of the selected biomarkers into diagnostic management. However, a major limitation is that these markers are not COVID-19-specific, and therefore, their application is meaningful only for monitoring purposes. Our analysis (Figure 2D) suggests that λFLC level is the best-suited candidate to distinguish between the two populations with an accuracy of 0.94 (0.87–1.00 95% CI), a precision of 0.91 (0.81–1.00 95% CI), a sensitivity of 1.00 (0.96–1.00 95% CI), and a specificity of 0.86 (0.70–1.00 95% CI). Additionally, we have some evidence that the combined use of λFLC and IgA2 might improve classification performance, but we do not have enough information to prove it statistically (Figure 3). A further study conducted on a larger sample size would help, on one hand, to confirm our results on λFLC, and, on the other hand, to decide whether a combined model would be more informative than the use of a single parameter. Unfortunately, the treatment of the latter issue is inherently complex as a strong correlation exists among all the investigated parameters, and between them and the blood levels of some inflammatory markers (Figure 4). This highly correlated behavior is not surprising as saliva FLC secretion has been shown to exhibit daily variation that reflects IgA fluctuation [5]. Despite the aforementioned complexity, a more in-depth evaluation of IgA2 would be highly interesting because of IgA2′s role and bio-distribution. IgA1 and IgA2 show different concentrations in serum and mucosal fluids [8]. While IgA1 strongly dominates in serum with an IgA1/IgA2 ratio of 9/1, in other biological fluids, including saliva, higher levels of IgA2 are reported [24,25]. A larger p-value is consistently detected in Figure 2 for IgA2 compared to IgA1, a finding that deserves a more in-depth study. The possible use of salivary IgAs, with an emphasis on IgA2, is even more interesting if one considers that the serological levels of total IgAs measured on 20/29 patients were within the range of normality (mean 2.32 g/L and standard deviation 0.92, data not shown). We stress here that the choice of measuring only 20 out of 29 patients was forced by the peculiar clinical path followed by each patient included in the study, as they were subject to pandemic restrictions. Interestingly, we also evaluated the Spearman’s correlation between salivary and blood levels in each patient, which showed significant correlations (*p* < 0.05) between total serum IgA and salivary IgA1 (ρ = 0.68), and between total serum IgA and total salivary IgA (ρ = 0.60).

Aside from IgAs, the large difference in salivary FLCs between the two groups is particularly intriguing as the immunological role of this molecular class is still not fully understood and is a matter of debate in the literature. Although salivary FLCs do not have antigen-binding specificity comparable to complete antibodies, they are thought to play a relevant role in the immune adaptive system [28]. In a previous paper, we hypothesized that FLCs could behave similarly to autoantibodies, shaping or skewing antibody reactivity [11,12]. Their active role in modulating the immune response is also supported by the overall expression of these molecules, only 60% of which are incorporated into whole Igs, with the remaining fraction released into serum as well as into other biological fluids [29]. Moreover, it is also possible that antigen specificity is not strictly necessary in salivary FLCs as it occurs in light chain-mediated hypersensitivity-like responses [30].

Interestingly, our Spearman’s coefficient map in Figure 4 does not highlight a significant correlation between FLC levels and age. This result is intriguing as other studies report significant age-related effects in saliva. For instance, Heaney et al. demonstrated that older adults show higher FLC salivary levels compared with young adults as a consequence of intense physical exercise [4]. In our opinion, the absence of any correlation with age in our dataset has two possible explanations: First, subjects included by Heaney et al. [4] span a larger age range, which goes from 18 to 80 years; second, and more interestingly, this result is likely to indicate that age-related effects are negligible in terms of immune activation in comparison with pathological states, particularly in COVID-19 patients.

The strong correlation existing between k and λ levels is not surprising and is likely to mimic the FLCs’ behavior in serum, where both light chains increase at the same time in response to inflammations/infections. Different behavior is often observed in other pathologies, such as cancer, where one chain type increases in excess compared to the other one. In this case, the k/λ ratio is a clinically reliable biomarker as it is not altered in nonspecific inflammatory processes. Consistently, we observe a modest and slightly significant difference in the k/λ ratio between the two groups (*p* = 0.013), further confirming that the FLC increase is likely to occur because of a nonspecific B-cell activation in the inflammatory process induced by SARS-CoV-2.

Taken together, our results can positively impact the development of effective protocols for monitoring SARS-CoV-2 patients in real time. In this context, it is worth stressing that molecular testing of nasopharyngeal swabs has some limitations: although necessary for diagnosis, it is time-consuming, and it requires specialized personnel as well as expensive equipment. Conversely, saliva is extremely easily accessible and can be sampled multiple times with no discomfort for the patients, thus providing an effective alternative for monitoring the disease and following up infected patients.

FLC levels outperform IgA in terms of performance in our database. This is likely related to the half-life of FLCs (a few hours) which is significantly shorter than IgA (21 days) [31]. Therefore, FLCs are likely able to provide an effective and real-time indicator of patient fluctuations, a particularly relevant issue in a pandemic era, as not all patients can access intensive care departments simultaneously. In this regard, it is worth stressing the strong and positive correlation between selected salivary biomarkers and D-Dimer levels in the blood, as this parameter was assessed as an effective biomarker for monitoring COVID-19 severity [32,33,34,35,36,37]

This study has some limitations: the number of patients studied was limited and the results of ongoing studies still have not provided evidence of immune protection of salivary IgA. However, saliva presents potential utility for monitoring local adaptive immunity.

In the pandemic era, tests of dynamic immune response that segregate patients from controls could be a sniper rifle for COVID-19 patients’ management.

## Figures and Tables

**Figure 1 jpm-11-00385-f001:**
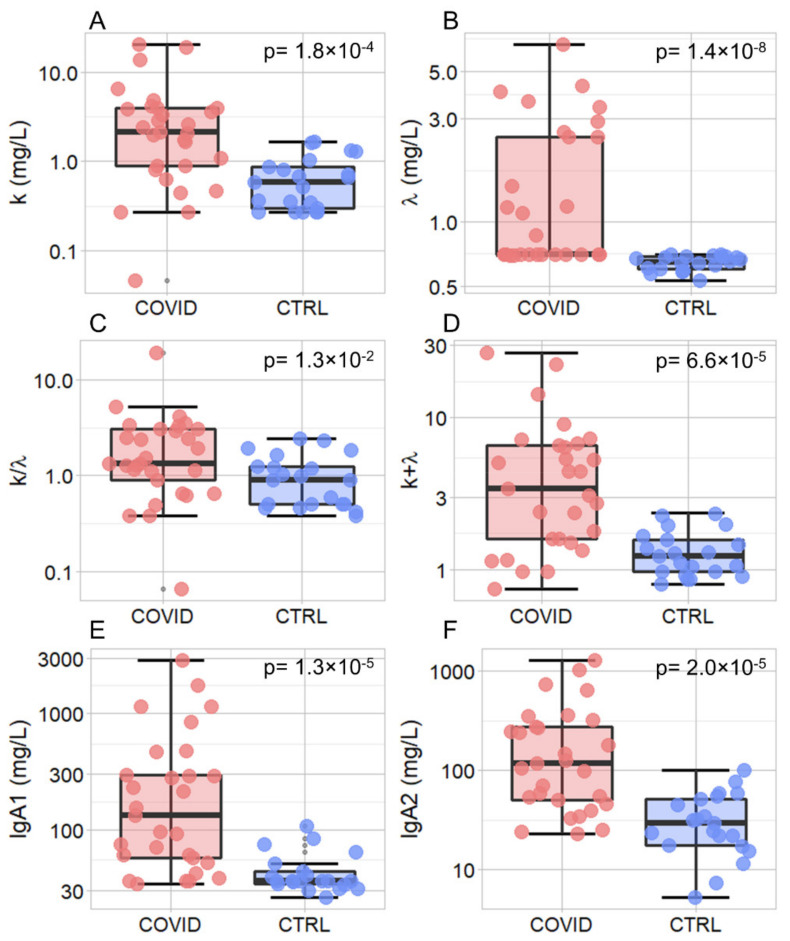
Box plot analysis of the salivary markers—k (**A**), λ (**B**), k/λ (**C**), k+ λ (**D**), IgA1 (**E**) and IgA2 (**F**)—measured in COVID-19 patients (light red) and control subjects (blue). Comparisons between the two groups were performed using the Wilcoxon Unpaired Two-Sample Test.

**Figure 2 jpm-11-00385-f002:**
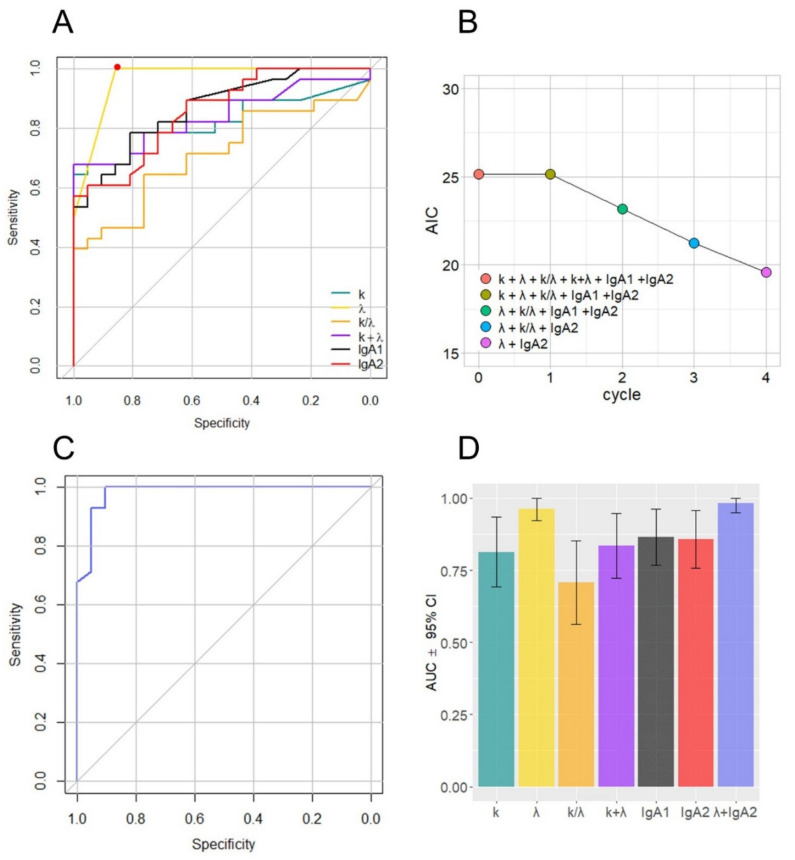
ROC curves for the selected salivary biomarkers (**A**); evolution of the Aikake Information Criterion during a stepwise logistic regression performed on all the measured parameters (**B**); ROC curve calculated using the selected variables (**C**); AUC values and 95% confidence intervals for all the computed ROC curves (**D**).

**Figure 3 jpm-11-00385-f003:**
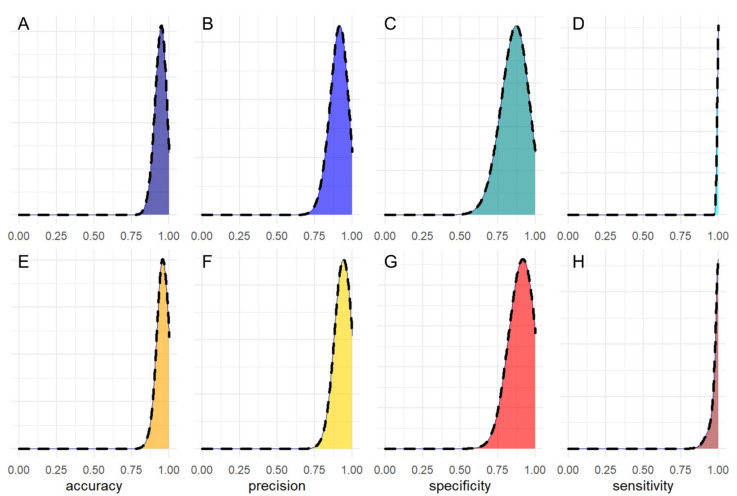
Bootstrap distribution of accuracy (**A**,**E**), precision (**B**,**F**), sensitivity (**C**,**G**) and specificity (**D**,**H**) for λ (upper panels) and λ + IgA2 (**E**–**H**).

**Figure 4 jpm-11-00385-f004:**
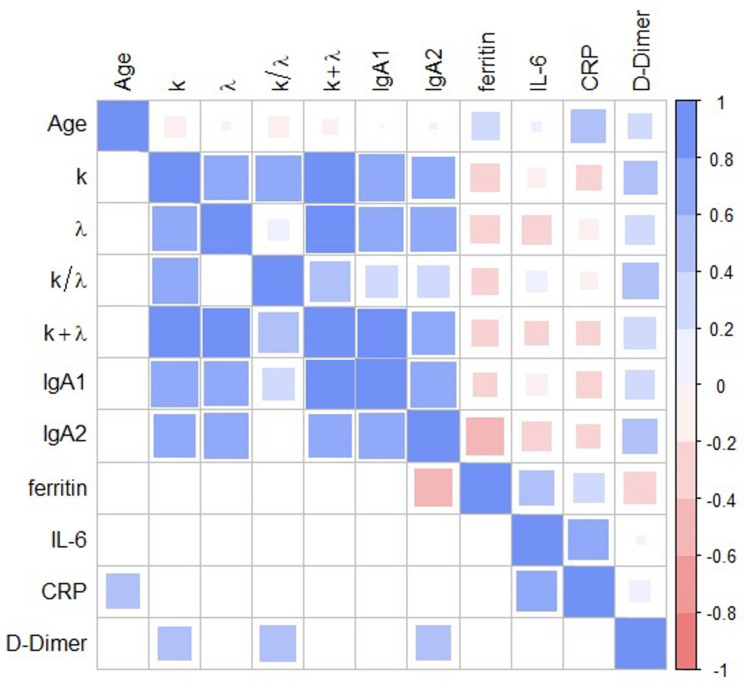
Heat map of Spearman’s correlation coefficients between salivary (FLCs and IgAs) and blood (ferritin, IL-6, CRP, D-Dimer) biomarkers. All the calculated correlations are plotted above the main diagonal and the significant ones below the main diagonal (α = 0.05).

**Table 1 jpm-11-00385-t001:** Clinical and demographic characteristics of the enrolled patients.

	Patients (*n* = 29)	Male (*n* = 17)	Female (*n* = 12)
**Age** (Mean ± SD)	49.7 ± 10.6	49.2 ± 13.6	50.3 ± 4.8
**Smoke** (%)	20.7	18.8	25.0
**Diabetes** (%)	10.7	12.5	8.3
**Hypertension** (%)	21.4	31.3	8.3
**Obesity** (%)	14.3	18.8	8.3
**Cardiopathy** (%)	14.3	18.8	8.3
**Oncology** History (%)	7.1	6.3	8.3
**Fever** (%)	78.6	81.3	75.0
**Cough** (%)	25.0	25.0	25.0
**Dyspnea** (%)	25.0	18.8	33.3
**Asthenia** (%)	26.0	37.5	33.3
**Pneumonia** (%)	75.0	81.3	66.7
**Oxygen Therapy** (%)	39.3	31.3	50.0

**Table 2 jpm-11-00385-t002:** Salivary and blood biomarkers in COVID-19 patients and healthy controls (CTRL). In column 2, we report reference values when available in literature.

		CTRL			Patients			
Analyte	Sample/ Reference Value	n	Median	IQR	n	Median	IQR	*p*-Value
k (mg/L)	Saliva	21	0.58	0.56	29	2.12	3.03	1.80 × 10^−04^
λ (mg/L)	Saliva	21	0.65	0.09	29	0.70	1.77	1.40 × 10^−^^08^
k/λ	Saliva	21	0.91	0.75	29	1.35	2.15	1.30 × 10^−02^
K + λ (mg/L)	Saliva	21	1.22	0.59	29	3.43	5.00	6.60 × 10^−^^05^
IgA1 (mg/L)	Saliva	21	36.30	9.70	29	133.10	237.10	1.30 × 10^−05^
IgA2 (mg/L)	Saliva	21	29.70	33.60	29	118.80	227.20	2.00 × 10^−05^
Ferritin (ng/mL)	Serum/ 30–300 ng/mL	−	−	−	27	432.00	286.00	−
IL-6 (ng/L)	Serum/ <7 ng/L	−	−	−	22	11.45	10.22	−
CRP (mg/L)	Serum/ 3–10 mg/L	−	−	−	25	28.60	42.50	−
D-Dimer (ng/mL)	Serum/ <250 ng/mL	−	−	−	28	437.50	393.75	−

**Table 3 jpm-11-00385-t003:** AUC values (95% CI) for ROC curves in Figure 2.

Parameter	AUC (95% CI)
k	0.81 (0.69–0.93)
λ	0.96 (0.92–1.00)
k/λ	0.70 (0.56–0.85)
k + λ	0.83 (0.72–0.94)
IgA1	0.86 (0.76–0.96)
IgA2	0.85 (0.75–0.95)
λ + IgA2	0.98 (0.95–1.00)

**Table 4 jpm-11-00385-t004:** Statistically significant Spearman’s correlation coefficients between salivary and blood biomarkers.

Groups	Coefficient	*p*
k vs. λ	7.16 × 10^−01^	1.25 × 10^−05^
k vs. IgA_1_	7.79 × 10^−01^	6.31 × 10^−07^
k vs. IgA_2_	6.33 × 10^−01^	2.22 × 10^−04^
k vs. D-Dimer	4.03 × 10^−01^	3.34 × 10^−02^
λ vs. IgA_1_	7.07 × 10^−01^	1.75 × 10^−05^
λ vs. IgA_2_	7.17 × 10^−01^	1.17 × 10^−05^
IgA_2_ vs. Ferritin	−4.76 × 10^−01^	1.20 × 10^−02^
IgA_2_ vs. D-Dimer	4.20 × 10^−01^	2.60 × 10^−02^
IL6 vs. CRP	6.01 × 10^−01^	5.02 × 10^−03^

## Data Availability

The data presented in this study are available on request from the corresponding author. The data are not publicly available as contain sensitive patient information.

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
