# Peer review of "Salivary Biomarkers in COVID-19 Patients: Towards a Wide-Scale Test for Monitoring Disease Activity"

_jpm, 2021, doi:10.3390/jpm11050385_

Round 1
Reviewer 1 Report
The search for biomarkers in COVID-19 in readily available biological material is an extremely important aspect of research in the era of an ongoing pandemic. Finding and identifying such compounds would allow a quick diagnosis or facilitate the monitoring of disease progression and the patient's condition. hence the study conducted by the Authors of the manuscript is very valuable. There are, however, some inaccuracies in the work. Below is a list of comments:
1. Basic demographic data for the control group are missing. The more that, as mentioned, IgA may differ depending on the age of the subjects
2. What was the procedure for collecting a saliva and blood sample. Was it a specific time? Were patients not consuming or drinking prior to collecting the saliva sample? If so, how long? Did the subjects rinse their mouths before placing the swab?
3. IgA subclasses were also determined in the serum?
4. Why was IgA tested in 20 out of 29 samples?
5. What are the standards for FLC for saliva - they are not included in the description. Likewise, IgA subclasses. Are the same values valid for the serum identified as correct?
6. On what basis was the strength of correlation established for the indicated values?
7. Figure S1 mentioned in the text is missing. Suplements are not included. Similarly, Table S1
8. Figure 4 - does it show relationships only for saliva?
9. Was the correlation between the same parameters measured in saliva and serum of the same patient checked? If so, have any trends been observed?
10. Was box plot analysis also performed on the patients' blood? Were the results similar to saliva?
11. The authors themselves mention that the described tests can complement PCR tests, but cannot replace them. It may be interesting to determine how the tested parameters will change during the course of the disease. Performing such tests would be a perfect complement to the theories set out about the possibility of monitoring the course of the disease by controlling the parameters tested
Author Response
Dear Editor,
We have received you decision letter and we have carefully each referee’s comment. First we want to thank the three referees for suggestions and criticisms, which helped us improving the quality and the readability of the paper.
Please find below our detailed answer (red) to each referee’s comment (black).
Dr.. Umberto Basile
(Guest editor of the special issue)
Reviewer 1
The search for biomarkers in COVID-19 in readily available biological material is an extremely important aspect of research in the era of an ongoing pandemic. Finding and identifying such compounds would allow a quick diagnosis or facilitate the monitoring of disease progression and the patient's condition. hence the study conducted by the Authors of the manuscript is very valuable.
We thank the reviewer for the understanding and the appreciation of our work.
There are, however, some inaccuracies in the work. Below is a list of comments:
Basic demographic data for the control group are missing. The more that, as mentioned, IgA may differ depending on the age of the subjects
We thank the referee for carefully reading the paper. We added this information in the section “patients and controls). We also pointed out the absence of statistically significant differences in the age between the two groups.
Specifically, the text was modified as follows:
We enrolled 29 COVID-19 patients (17 males and 12 females, mean age 49.7 years, SD 10.6 years) for this monocentric prospective study among patients admitted to COVID-19 Unit at Fondazione Policlinico universitario “A.Gemelli”, IRCCS, Rome; 21 healthy donators (HD) (10 Male and 11 Female, mean age 44.7 years, SD 11.6 years ) served as control group. Age values between the two groups were compared with a Wilcoxon Test (p=0.19 ), pointing out the absence of statistically significant differences.
What was the procedure for collecting a saliva and blood sample. Was it a specific time? Were patients not consuming or drinking prior to collecting the saliva sample? If so, how long? Did the subjects rinse their mouths before placing the swab?
We admit that the required information were poorly written in the previous version of our paper, therefore we thank the reviewer for highlighting this weakness of our paper.
Blood was collected by venepuncture into plain vacutainer tubes. Whole blood samples were allowed to clot at room temperature for 20min and then centrifuged at 3000×g for 10min at 4°C; separated serum was immediately stored at −80°C until analysis. Unstimulated saliva samples were collected with participants in the seated position, leaning forward with their head tilted forward. For saliva sampling, patients and healty controls were provided with the following instructions before samples were taken: avoid the consumption of solid foods one hour before the test and perform oral and dental hygiene half an hour before the evaluation. Sampling was performed between 09:00 h and 11:00 h to minimize the variations associated with the circadian cycle. Saliva collections lasted 3min, during which time participants had a salivette in mouth side theet, subsequently centrifuged at 10,000×g for 2min to separate cells and insoluble matter. The supernatant was then removed and stored at −80°C until assay.
IgA subclasses were also determined in the serum?
Unfortunately, we can’t provide these values for the recruited subjects. We agree with the referee that a more in-depth comparison between subclasses would be extremely interesting to further validate the studied salivary markers, which could be the subject of a next study.
Why was IgA tested in 20 out of 29 samples?
This was an unavoidable choice associated to the different clinical path of each patient during pandemic restrictions. We comment on this point in the discussion section. We comment on this point in the discussion section
What are the standards for FLC for saliva - they are not included in the description. Likewise, IgA subclasses. Are the same values valid for the serum identified as correct?
We added salivary reference values in table 2. Serum normal ranges were reported in section “laboratory testing”
On what basis was the strength of correlation established for the indicated values?
We chose the classification reported in Taylor, Richard. "Interpretation of the correlation coefficient: a basic review." Journal of diagnostic medical sonography 6.1 (1990): 35. A reference was added in the statistical section
Figure S1 mentioned in the text is missing. Suplements are not included. Similarly, Table S1
The Authors gratefully acknowledge the reviewer for carefully reading the paper. Supplementary Materials, which include Figure S1 and Table S1, are provided in the revised version of the manuscript in a separate file (supplementary materials.doc).
Figure 4 - does it show relationships only for saliva?
We thank the referee for the comment. We admit this point was poorly specified in the previous version of the paper. Specifically, CRP, ferritin, IL6, D-dimer are measured in blood. In the revised version of the paper we clearly state whether the parameter is measured on blood or saliva in table 2. We further specified this information in the caption of figure 4.
Was the correlation between the same parameters measured in saliva and serum of the same patient checked? If so, have any trends been observed?
We thank the referee for the suggestion. We checked for correlations between the some parameters in blood and saliva. Correlation was evaluated using the Spearman coefficient. We found two significant correlation (p<0.05) between total serum IgA and salivary IgA1(ρ=0.68) and total salivary IgA (ρ=0.60). We comment on this point in the discussion section .
Was box plot analysis also performed on the patients' blood? Were the results similar to saliva?
Unfortunately, we are not able to provide all the serum data on the control group and so we didn’t perform the comparison of blood levels in the two groups.
The authors themselves mention that the described tests can complement PCR tests, but cannot replace them. It may be interesting to determine how the tested parameters will change during the course of the disease. Performing such tests would be a perfect complement to the theories set out about the possibility of monitoring the course of the disease by controlling the parameters tested
We completely agree with the referee and we thank him/her for the suggestion: tested parameters should have been determined also during the course of the disease for monitoring purposes. We measured these salivary levels during the first pandemic outbreak in Italy between may and july 2020. Therefore, we are not able anymore to follow up the course of the pathology, which could be the focus of a next paper. We commented on this point in the discussion section.
Reviewer 2 Report
This is an interesting study and timely.
What could be a possible reason for the observed decrease k/λ ratio in patients with diabetes and hypertension as observed by the authors?
Why did the authors decide to use AIC vs BIC in choosing a model?
Author Response
Dear Editor,
We have received you decision letter and we have carefully each referee’s comment. First we want to thank the three referees for suggestions and criticisms, which helped us improving the quality and the readability of the paper.
Please find below our detailed answer (red) to each referee’s comment (black).
Dr.. Umberto Basile
(Guest editor of the special issue)
This is an interesting study and timely.
We thank the referee for the appreciation of our work.
What could be a possible reason for the observed decrease k/λ ratio in patients with diabetes and hypertension as observed by the authors?
We agree with the referee, and we thank her/him for stimulating us to provide a more in-depth discussion on this interesting issue. Attempting to answer to the referee’s question, we noticed that our results are in close agreement with those recently published by Matsumori and collaborators (Infammation Research (2020) 69:715–718) which measured a decrease in the k/λ ratio on blood samples obtained from a cohort of diabetic subjects compared to control ones. The authors hypothesize an involvement of NF-kB activation, which might influence differently the production of FLC kappa and FLC lambda. Concerning hypertension, we noticed - as expected - a strong association between diabetes and hypertension in our dataset. Therefore, we suppose that the decrease in the kappa/lambda ratio in the two groups might be correlated. We commented on this point in the result section.
Why did the authors decide to use AIC vs BIC in choosing a model?
We thank the referee for this question. We tested both methods and we obtained the same selected variables. We commented on this point in the results section.
Reviewer 3 Report
I Carefully read the manuscript and I have some comments/suggestions for authors
The text needs some improvement in English writing.
ABSTRACT
Q1) the introduction is quite long. The methods and results sections need to be reworded.
MATERIAL AND METHODS
Q2) The exclusion criteria of the study are not very clear. Could the authors better explain exclusion criteria to health controls and other exclusion criteria to enrolled patients? (for example autoinmnune diseases and other diseases with elevated levels of Ig A, primary and adquired immunodeficiency etc..)
Q3) Are you verified other previous pathologies in enrolled patients?
Q4) The authors must be included Methodology of all parameters included in the manuscript as well as the collection and preservation of serum samples
Q5) Table 1: Could be of interest, carry out the table with the data of enrolled patients differentiate by sex
Q6) Table 2. This table must be reestructured. It is necessary to include for each analyte, sample type (serum or saliva) and the reference values. It would be the interest to add IgA serum results and their reference values also. You must explain the interest to describe serum results for Ferritin, IL-6, PCR and D-Dimer only in enrrolled patients. This results not are well explained throughout results or discussion.
RESULTS
Q7) The results appear sometimes somewhat confusing in the text. On the other hand, there are some sentences that fit better in material and methods Section. Authors should better clarify this information. Definitely, this part needs to be formulated more precisely and clearly for the reader
Q8) Figure 2: The descriptions in Figure 2 show several mistakes. Please Check it.
Q9) Figures (1, 2,3 and 4) The poor quality and the small size of the figures make them difficult to see the results.
Author Response
Dear Editor,
We have received you decision letter and we have carefully each referee’s comment. First we want to thank the three referees for suggestions and criticisms, which helped us improving the quality and the readability of the paper.
Please find below our detailed answer (red) to each referee’s comment (black).
Dr.. Umberto Basile
(Guest editor of the special issue)
I Carefully read the manuscript and I have some comments/suggestions for authors
The text needs some improvement in English writing.
The text was edited under the supervision of a native English speaker.
ABSTRACT
Q1) the introduction is quite long. The methods and results sections need to be reworded.
We shortened the introduction from 670 words to 540 words. Results were modified according to the comment Q7 of this referee. Please see our detailed answer to this comment. Methods were modified according to question Q2, Q4 and Q7 of this referee. Please see our detailed answers to these comments.
MATERIAL AND METHODS
Q2) The exclusion criteria of the study are not very clear. Could the authors better explain exclusion criteria to health controls and other exclusion criteria to enrolled patients? (for example autoinmnune diseases and other diseases with elevated levels of Ig A, primary and adquired immunodeficiency etc..)
We admit the this point was not clearly explained in the previous version of our manuscript and we thank the referee for carefully reading the paper highlighting possible weaknesses and unclear points. In the revised version of the paper, we commented more in-depth on this specific point in the section patients and controls.
Patients and subjects under the age of 18 were not included. Other exclusion criteria were the presence of plasma cells disorders, autoimmune disease, the presence of autoantibody, anti-nuclear antibody positivity, the presence of a monoclonal component in capyllaris electrophoresys, presence of chronic hepatitis (C,B).
Q3) Are you verified other previous pathologies in enrolled patients?
The patients enrolled in this study do not present other previous pathologies except for those listed in their clinical history (table 1 and table S1).
Q4) The authors must be included Methodology of all parameters included in the manuscript as well as the collection and preservation of serum samples
Again we thank the referee for carefully reading the paper. We commented more in depth on this points in the laboratory testing section. Specifically, the following text was added to the paragraph:
Blood was collected by venepuncture into plain vacutainer tubes. Whole blood samples were al-lowed to clot at room temperature for 20 min and then centrifuged at 3000×g for 10min at 4°C; separated serum was immediately stored at −80°C until analysis. Unstimulated saliva samples were collected with participants in the seated position, leaning for-ward with their head tilted forward. For saliva sampling, patients and healthy controls were provid-ed with the following instructions before samples were taken: avoid the consumption of solid foods one hour before the test and perform oral and dental hygiene half an hour before the evaluation. Sampling was performed between 09:00 h and 11:00 h to minimize the variations associated with the circadian cycle. Saliva collections lasted 3min, during which time participants had a salivette in mouth side theet, subsequently centrifuged at 10,000×g for 2min to separate cells and insoluble matter. The supernatant was then removed and stored at −80°C until assay.
Q5) Table 1: Could be of interest, carry out the table with the data of enrolled patients differentiate by sex
Table 1 was restructured as suggested by the reviewer. A new version of the table, which include the data of enrolled patients differentiate by sex, is provided with the revised version of the manuscript.
Q6) Table 2. This table must be restructured. It is necessary to include for each analyte, sample type (serum or saliva) and the reference values. It would be the interest to add IgA serum results and their reference values also.
Table 2 was modified according to the referee’s suggestions.
You must explain the interest to describe serum results for Ferritin, IL-6, PCR and D-Dimer only in enrrolled patients. This results not are well explained throughout results or discussion.
We agree with the referee, this is an important point and deserve a more extended discussion. We decided to investigate these parameters as their alteration has been often reported in the literature. We added several references on the subject and we explaind more clearly the rational behind their investigation in the present paper. Interestingly, we found a significant correlation between k and (and as a consequences, k/λ levels) and IgA2 and D-dimer. D-dimer, in its turn, has been D-dimer and IgA2 and
RESULTS
Q7) The results appear sometimes somewhat confusing in the text. On the other hand, there are some sentences that fit better in material and methods Section. Authors should better clarify this information. Definitely, this part needs to be formulated more precisely and clearly for the reader
We agree with the referee that part of the result section fits better into material and methods. Therefore, the following parts were moved from results to methods:
- ROC curve explanation and how axes are visualized.
- AIC definition
- Bootstrap implementation for the two classification models.
- description of how the correlation map s visualized in terms of significant correlation (figure 4) was moved to the caption of figure 4
We want to thank the referee for this suggestion, which we believe helped us to greatly improve the paper readability.
Q8) Figure 2: The descriptions in Figure 2 show several mistakes. Please Check it.
We thank the referee for carefully reading the paper. We carefully checked for mistakes in the figures
Q9) Figures (1, 2,3 and 4) The poor quality and the small size of the figures make them difficult to see the results.
Figures were resized and are clearly presented in the revised version of the manuscript. High definition figure, were also uploaded in the revised version of the paper.
Round 2
Reviewer 3 Report
Your manuscript addresses a topic of great interest and utility. After reviewing your second version , I consider that the manuscript has improved significantly